# Knowledge, Compliance, and Inequities in Colon Cancer Screening in Spain: An Exploratory Study

**DOI:** 10.3390/healthcare11182475

**Published:** 2023-09-06

**Authors:** Mario López Salas, Diego De Haro Gázquez, Belén Fernández Sánchez, María Luz Amador Muñoz

**Affiliations:** Asociación Española Contra el Cáncer, Teniente Coronel Noreña, 30, 28045 Madrid, Spain; diego.deharo@contraelcancer.es (D.D.H.G.); belen.fernandez@contraelcancer.es (B.F.S.); mluz.amador@contraelcancer.es (M.L.A.M.)

**Keywords:** FOBT, screening, colorectal cancer, early detection

## Abstract

In Spain, inequities exist in implementing colorectal cancer (CRC) tests with the target population—adults aged 50 to 69—as part of population-based CRC screening programs. This research aims to further our understanding of the target population’s awareness, attitudes, and perceptions of these test-based screening programs. A survey was carried out using an online panel representative of the target population, with a sample collected from 5313 individuals. Data collection took place in June 2022. Descriptive and bivariate analyses were carried out using contingency tables, the Chi-square test, and Cramer’s V statistics. The sample was also segmented based on key variables. Finally, the results were analyzed using logistic regression. In the sample population, 62.5% had taken the fecal occult blood test (FOBT), 72.5% reported receiving the invitation letter to participate in the screening program, and 86.8% had prior knowledge of the FOBT. Noncompliance was mainly due to lack of symptoms (40%), non-receipt of invitation letters (39.7%), and forgetfulness or neglect (28.5%). On the contrary, receipt of the letter of invitation (OR 7.35, *p <* 0.01) and prior knowledge of FOBT (OR 6.32, *p <* 0.01) were the main variables that increased the probability of test uptake. Other significant variables included frequency of primary care visits (OR 1.71, *p <* 0.01) and being older (65–69 years old) (OR 1.52, *p <* 0.01) There is still a pressing need for greater awareness of both CRC risk factors and the benefits of early detection, as well as for overcoming the common misconception that detection should only be sought when symptoms are present.

## 1. Introduction

Colorectal cancer (CRC) is a major public health issue with high incidence and mortality rates. In Spain, it is estimated that 41,646 new cases of CRC were detected in 2022, making it the country’s most frequently diagnosed tumor. In terms of gender, it ranks second in frequency in both men (25,406) and women (16,240) [1].

Several screening tests are currently available for the early detection of CRC, with strong evidence supporting the efficacy of the fecal occult blood test (FOBT) and sigmoidoscopy [2,3,4,5]. In Spain, CRC screening began in the late 1990s and has gradually grown through pilot programs in different regions [6]. The first population-based screening for CRC was carried out as a pilot program in Catalonia in 2000 [7]. In 2009, it became an objective of the National Health System (NHS) Cancer Strategy to implement and extend population-based screening for CRC. This initiative targeted moderate-to-low-risk individuals aged 50–69 for biennial screening using the FOBT, followed by colonoscopy in positive cases [8]. This strategy was endorsed in 2013 after the NHS Interterritorial Council agreed to include this type of screening in the NHS portfolio of standard services. These agreements were published in the Official Bulletin of the Spanish State (BOE) [9] with effective entry into force in 2014.

It is estimated that population-based screening programs must achieve participation rates of over 60% to be successful and reduce CRC mortality with adequate cost-effectiveness [10]. The European Guidelines for Quality Assurance in CRC Screening and Diagnosis recommend a minimum participation rate of 45%, although optimal rates should exceed 65–70% [11]. In Spain, the NHS Cancer Strategy recommends a minimum participation rate of 65% in CRC screening programs by the target population [12].

However, current data indicate that Spain still needs to achieve this level of participation. According to the Network of Cancer Screening Programs, coverage reached 44% of the target population in 2017, with 46% participation and high variability among autonomous communities (19–74%) [13]. In Europe, accurate comparisons of CRC screening compliance between countries are challenging, given the differences in data updating, target age range, detection intervals, and choice of tests for screenings [14]. However, a review by Navarro et al. reveals a high variation in participation rates throughout Europe, with the highest compliance in the Netherlands and the lowest in the Czech Republic [15]. Nevertheless, the research indicates that screening programs should employ specific strategies to engage the target population and encourage participation.

Since the publication of the first CRC screening guidelines, much attention has been focused on researching factors associated with CRC screening participation. Research in the literature has identified numerous demographic, socioeconomic, and attitudinal variables influencing target population compliance with FOBT-based screening, the most significant being gender, age, ethnicity, education level, and socioeconomic status [6,16,17,18,19,20]. The data show that women and older individuals have the highest rates of participation [21,22]. A study carried out in the Basque Country, in which the overall participation rate (64.3%) was close to the recommended 65%, showed lower rates in men compared to women and in the youngest and oldest age groups [23].

The influence of socioeconomic status on participation varies between genders, with less participation from men with lower socioeconomic status (60.3%) and women who occupy higher socioeconomic positions (65.7%) [23]. The findings of Frederiksen and colleagues on a sample of 177,114 individuals revealed that low socioeconomic status, as measured by education level, employment, and income, was strongly associated with low screening prevalence. There is solid evidence indicating the existence of social inequalities in CRC screening, but few studies analyze the root causes of these inequalities. Some publications agree that the main reasons for nonparticipation among the most disadvantaged socioeconomic groups are a need for more awareness of the disease, prioritization of issues with a more significant impact on daily life, and the inability to comprehend written invitations to the program [6,18,19,24].

While the barriers to FOBT implementation are well described in the literature, the problem of engaging more people, particularly nonparticipants, in screening programs still needs to be solved. Few studies have explored strategies aimed at rescuing nonparticipants and identifying the motives for the noncompliance of this often sizeable portion of the population [21]. As suggested by Navarro and colleagues [15], if the goal is to design strategies that promote equitable access to CRC screening programs, we must first achieve a better understanding of the predictors and barriers to program participation.

This study aims to explore the perceptions, beliefs, and attitudes which act as emotional and rational factors for CRC screening participation or nonparticipation to analyze the barriers noted by nonparticipants and identify elements for improvement.

## 2. Materials and Methods

### 2.1. Research Design

We used an exploratory quantitative cross-sectional survey design to meet the research objectives.

### 2.2. Participants and Sample

The target population for the CRC screening program, men and women aged 50 to 69 residing in Spain, was eligible to participate in the study. The participants were selected through a web-based survey among subjects distributed throughout the entire Spanish territory who were members of an online respondent Kantar panel. As a result, the sample of this study was composed of 5313 participants. Data collection was carried out in June 2022. The maximum statistical error of the overall data was +/−1.3%, under simple random sampling standards with a confidence interval of 95.5%.

### 2.3. Measures

An ad hoc questionnaire was designed based on the results of a previous qualitative study using four discussion groups [25] and bibliographical reviews of analogous research that analyzed variables associated with participation in CRC screening programs. The content of the questionnaire was reviewed and validated by a group of experts and divided into different sections as follows. The questionnaire is available as Appendix A.

#### 2.3.1. Health Status and Lifestyle

Information was collected on self-reported health status and self-perception of lifestyle regarding their health.

#### 2.3.2. Health Care and Attitudes towards Check-Ups

This section gathered data on the types of health services the respondents had received (i.e., frequency of primary care visits, medical check-ups, and tests) and personal perceptions of generic medical check-ups and tests and, more specifically, of tests related to cancer prevention, and the importance given to them.

#### 2.3.3. Colorectal Cancer: Knowledge and Risks

Respondents were asked about prior knowledge of colorectal cancer, the self-perceived lifetime risk for CRC, and reasons for high- or low-risk perception. Information was also gathered on beliefs and opinions regarding the epidemiology of CRC and perceived incidence as a function of gender and age.

#### 2.3.4. CRC Screening Program and FOBT Uptake

This section asked respondents if they had prior knowledge of the CRC Early Detection Program and the fecal occult blood test (FOBT), if they had received the invitation letter to participate in this program by providing an FOBT sample, and if they had or had not carried out the FOBT and why. Respondents who underwent the test answered questions on general attitudes and perceptions regarding the FOBT. Those who had not received the invitation letter or had no prior knowledge of the test were asked about the possibility of carrying out this test and the reasons for doing so. Finally, the questionnaire requested an overall assessment of the CRC Early Detection Program and the information available to the general public about its existence and benefits.

### 2.4. Data Analysis

A descriptive analysis of the relevant variables was carried out using frequency and percentage tables. Bivariate analyses were performed for the between-group comparison of socio-demographic variables, using contingency tables, the Chi-square test, and Cramer’s V statistic to detect relationships and their effect size. The sample was also segmented according to age and other variables, including high or low self-perceived risk for CRC and FOBT uptake. Finally, a logistic regression model was carried out to determine which variables increased the probability of FOBT uptake. For the construction of the model, maximum likelihood estimation and backward stepwise regression were used. The model yielded a Nagelkerke coefficient of determination of R^2^ = 0.397, which is considered acceptable.

For evaluating the statistical significance of the findings in this investigation, we set a significance level of <0.05. Data were analyzed using IBM SPSS Statistic v27 [26].

## 3. Results

### 3.1. Participants’ Demographic Characteristics

A total of 5313 participants were included in this study. Table 1 shows the sample distribution according to the main socio-demographic variables.

### 3.2. Health Status, Health Care, and Attitudes toward Check-Ups

A total of 68.3% of the sample rated their health status as good or excellent. Regarding lifestyle, 84.7% perceived themselves as having a healthy or very healthy lifestyle, and 88.2% of the sample visited their physician at least once a year (Table 2).

Over 90% of the respondents (91.5%) had a positive perception of check-ups, agreeing that it is advisable to conduct medical tests even if a person is healthy and asymptomatic. Regarding the performance of specific tests as the best option for the early detection and treatment of certain types of cancer, 96.8% strongly agreed or agreed. In total, 43.7% strongly agreed or agreed with the statement that the onset of cancer is primarily random or due to genetics, and little can be done to prevent it. Finally, 85.2% agreed that a healthy lifestyle could prevent the onset of cancer, and 91.9% held that ongoing follow-up with their physician helps the early detection or prevention of cancer.

### 3.3. Colorectal Cancer: Knowledge and Risks

In total, 97.6% of the participants had heard or known about colorectal cancer before the survey. Furthermore, 27.9% considered themselves as having a high or very high risk of CRC in their lifetime. Table 3 shows the primary factors cited by the respondents for perceived risk for CRC.

Regarding beliefs and attitudes on CRC, 83.1% of the sample believed that this disease had a high or very high incidence rate in the general population when it is compared to other forms of cancer. In terms of gender, 52.4% of the participants considered that CRC affects more males than females, whereas 43.3% of the sample considered it affects both genders equally, and only 2.3% believed that CRC incidence is higher among women.

### 3.4. CRC Screening Program and FOBT

Our research showed that 86.8% of respondents had prior knowledge of the FOBT used in Spain for CRC screening. Moreover, 72.5% received the invitation letter sent by public institutions to promote screening participation by providing an FOBT sample. Of these, 78.9% knew the test before receiving the letter. A total of 62.5% complied with the test. Table 4 shows the main reasons given for FOBT uptake or non-uptake.

The results about perceptions and attitudes related to FOBT among the sample of this study showed a high degree of consensus on the benefits of performing this test. Thus, almost all participants (99.3%) thought that the advantages were far more important than the damages or inconveniences of carrying it out. In addition, 99.6% of the participants who had undergone this test before did so because they considered that it was an effective measure to take care of their health, and 99.8% agreed that this type of test aimed to achieve the early detection of possible cancer which is essential to improve the prognosis and subsequent treatment. Furthermore, 98.9% stated that the FOBT is simple and easy to perform.

Of those who did not take the FOBT, 67.1% agreed or strongly agreed that a negative test does not exempt the possibility of developing CRC in the future. In total, 51.5% affirmed that uncertainty regarding the test result could be distressing; 36.5% agreed or strongly agreed that the test is uncomfortable and unpleasant to perform; and 32.8% believed that carrying out the FOBT is unnecessary when the person is asymptomatic and in good health. Regarding future expectations for FOBT compliance, 75% stated they would take the test in the future.

Finally, in the overall assessment of the CRC Early Detection Program, 97.1% had a positive or very positive opinion of the program. However, only 26.3% affirmed that information available to the general public regarding the program is sufficient or adequate, while 68.6% deemed it needed to be more adequate, insufficient, or null.

### 3.5. FOBT Uptake: Bivariate Analysis

Bivariate analyses between the receipt of the invitation letter and prior knowledge of the test on the likelihood of FOBT uptake showed that 77.1% of the individuals who received the letter underwent the test. As shown in Table 5, respondents with prior knowledge of FOBT reached 69.9% in screening compliance. In addition, significant differences with moderate effect size were found between attitudinal variables and self-perceived risk on FOBT uptake. These variables include the frequency of primary care visits, the perception of routine medical check-ups, and the perception of the effectiveness of cancer screening tests.

Regarding socio-demographic variables, age proved to be the only variable with significant differences and moderate effect size on FOBT uptake.

### 3.6. Logistic Regression Model: FOBT Uptake

Table 6 displays the results of the logistic regression model with effective FOBT implementation as the dependent variable. The variables which increased the probability of this outcome the most were the receipt of the invitation letter (OR 7.35) and prior knowledge of FOBT (OR 6.32). Moreover, people who visited their primary care physician every six months or more frequently (OR 1.71) and had a positive perception of routine medical check-ups (OR 1.69) had a higher probability of FOBT screening compliance. The older age groups within the target population were more likely to take the test than the younger age groups (OR 1.52), as were subjects with self-perceived risk for CRC (OR 1.41). Finally, place of residence also contributed significantly to FOBT uptake, highlighting the vast heterogeneity among the geographic regions of Spain.

## 4. Discussion

Participation is a crucial indicator of the effectiveness of CRC screening programs. A high participation rate among the target population is also needed to significantly reduce mortality [27]. Nevertheless, our research in Spain, which corroborates findings in other countries, reveals numerous barriers to participation in CRC screening programs.

First, our data show that citizens rated their health status and lifestyle quite highly: 7 out of 10 respondents had a positive view of their health, and 84.7% believed they led a healthy lifestyle. Similar studies, however, have not found a clear association between perceived health and screening compliance: some of them showed that poor general perception of health is a barrier to participation [28,29,30]. In contrast, other research identified it as a facilitator [31]. In our study, neither self-perceived health nor lifestyle assessment was associated with FOBT uptake. This may be because most individuals who have participated in our study exhibited a highly positive perception regarding the healthiness of their lifestyle. This may result in a high level of “optimism bias,” a phenomenon in which some people believe they have a lower risk of a specific danger than others. In addition, it may reinforce the notion that early detection testing can be postponed. The positive assessment of one’s health status can lead to the perception that the disease is distant, thereby lacking a direct sense of urgency and ultimately leading to the postponement of FOBT performance. These groups are unlikely to be further motivated solely based on general health recommendations. Even so, it would be of interest to take advantage of the ever-growing concern for health and self-care and establish preventive testing in general—and FOBT-based screening in particular—as a fundamental and beneficial means to promote one’s health.

Regarding the utilization of health care services, 90% of the respondents reported at least yearly visits to primary care physicians. When analyzing the relationship between the frequency of visits to primary care and FOBT uptake, we found that more visits to the doctor increased the likelihood of screening participation. These results are consistent with previous studies [32,33,34] showing that less contact with healthcare providers lowered screening compliance while receiving information from physicians on early detection encouraged participation. The perception of the effectiveness of cancer screening tests has also been linked to increased testing. Thus, it would be advantageous to use primary care visits to inform patients and enhance the role of health professionals whose expert advice could help legitimize the test.

Regarding CRC awareness and perceptions of the disease, almost the entire survey population was familiar with this type of cancer (97.6%). Nevertheless, only 27.9% reported a high or very high self-perceived lifetime risk for CRC, compared to 69.5%, who viewed their risk as low or very low. This finding is of great significance when it comes to formulating educational recommendations aimed at increasing the number of individuals performing an FOBT. In this context, we delved into the health belief model (HBM) and the protection motivation theory (PMT) [35,36,37], highlighting the crucial role of each individual’s subjective perception of their vulnerability to illness in predicting health-promoting and health-protective behaviors. According to the model, it becomes evident that individuals are more likely to consider changing their behaviors when they personally perceive a threat posed by a specific disease or ailment. It is only at this point that they engage in a personal cost–benefit analysis of the proposed behavior change, evaluating the personal costs involved and the benefits they stand to gain. In addition, other novel and interesting data from our study has been to determine the main reasons for reporting a high CRC risk perception were attitudinal, while factors related to leading a healthy lifestyle were secondary. Hence, this finding suggests that we must improve education on CRC risk factors, especially those underestimated when assessing personal risk, including sedentary lifestyles, obesity, poor diet, smoking, and alcohol consumption.

Familiarity with the CRC screening program and the FOBT was high: 82.4% had prior knowledge or had heard of the screening program, and 86.9% were familiar with the FOBT. Furthermore, having knowledge of the test should be considered a facilitating factor for CRC screening compliance. Our results are consistent with prior research showing that participation in CRC screening programs is higher in individuals with greater knowledge of the test [21,38].

Regarding FOBT uptake, 62.5% of the respondents had carried out the FOBT. Participation rose to 77.1% if the respondents had received the CRC screening invitation letter from health authorities in Spain. Our data confirm that receipt of the invitation letter to carry out the FOBT, sent by health authorities in the context of the colorectal cancer screening program in Spain, is the variable that most significantly increased the probability of test implementation. Furthermore, residing in one region vs. another may increase or decrease this probability, positively highlighting regions where the program has universal coverage and a longer period of development compared to regions where the program has been implemented more recently and has not reached coverage rates for the entire target population.

Analysis of the attitudinal variables of participants revealed that ruling out the possibility of having the disease was the primary reason for taking the FOBT, followed by the belief that early detection leads to effective treatment in most cases. The third most reported reason for compliance was being in the age group recommended for CRC screening. We also found a predominance of arguments supporting the ease and simplicity of the overall test process.

Concerning nonparticipation, our results were consistent with previous studies that cited a lack of awareness, the absence of symptoms, low CRC risk perception, and forgetfulness or neglect as the primary reasons for noncompliance [29,33,34,38]. Age was also a critical variable, given that younger respondents had a lower sense of vulnerability to disease, making this a barrier to participation [17]. In sum, we found that FOBT uptake is not associated with negative attitudes towards the screening program or the test but has mainly to do with (1) receipt of the invitation letter to participate in the program; (2) viewing colon cancer as a potential health concern; (3) awareness of the importance of early detection; and (4) socio-demographic characteristics (such as age).

This study has some limitations. Firstly, although our sample was national, it was not a randomized sample, and therefore it may not fully reflect the entire Spanish population aged 50–69 years old. Secondly, given the online modality of the survey, the sample may be skewed in favor of individuals with access to the Internet. Thirdly, as we relied on self-reported measures, the results might have been influenced by self-report bias, such as selective memory, social desirability, or attribution bias, which refer to the act of attributing positive events and results to one’s person but attributing negative events and results to external forces.

## 5. Conclusions

Expanding the scope of population-based screening programs remains a priority, especially in countries such as Spain, where full coverage of the target population has yet to be achieved. However, CRC screening compliance is influenced by awareness and attitudinal factors, which can limit participation and encumber effective CRC program implementation. These factors must be considered in efforts to provide education on CRC risk factors and the benefits of early detection and when addressing the common misconception that detection should only be sought when symptoms are present.

## Figures and Tables

**Table 1 healthcare-11-02475-t001:** Socio-demographic characteristics of the sample (*n* = 5313).

Variables	% (*n*)
**Age**	
50–54	29.3 (1557)
55–59	27.1 (1440)
60–64	23.9 (1270)
65–69	19.7 (1047)
**Gender**	
Male	48.9 (2598)
Female	51.1 (2715)
**Education Level**	
Primary education or less	18.4 (978)
High School or less	43.0 (2285)
Higher education	38.6 (2051)
**Place of residence**	
Large cities (over 400,000 inhab.)	19.4 (1033)
Urban (50,001 to 400,000 inhab.)	32.0 (1701)
Semi-urban (10,001 to 50,000 inhab.)	27.2 (1446)
Rural (up to 10,000 inhab.)	21.3 (1133)

**Table 2 healthcare-11-02475-t002:** Distribution of variables on health status, health care, and attitudes toward check-ups.

Variables	% (*n*)
**Health status**	
Excellent	13.0 (691)
Good	55.3 (2938)
Fair	25.4 (1350)
Poor	5.0 (266)
Bad	1.3 (69)
**Lifestyle in terms of health**	
Very healthy	8.5 (452)
Quite healthy	76.2 (4049)
Slightly healthy	14.9 (792)
Not healthy at all	0.4 (21)
**Frequency of primary care visits**	
Once or several times a month	5.7 (303)
Every two or three months	21.9 (1164)
Every six months	31.4 (1668)
Once a year	29.2 (1551)
Once every few years	9.6 (510)
Never	2.2 (117)
**Perception of medical tests and check-ups**	
It is not necessary to perform these tests if a person is healthy	8.5 (452)
Even if a person is healthy, performing these tests is always advised	91.5 (4861)
**Conducting specific tests for certain types of cancer is the best option for early detection and treatment**	
Strongly agree	57.9 (3076)
Agree	38.9 (2067)
Disagree	2.8 (149)
Strongly disagree	0.4 (21)
**The onset of cancer is primarily random or due to genetics, and little can be done** **to prevent it.**	
Strongly agree	9.8 (521)
Agree	33.8 (1796)
Disagree	43.7 (2322)
Strongly disagree	12.7 (675)
**A healthy lifestyle can prevent the onset of cancer**	
Strongly agree	37.2 (1976)
Agree	48.0 (2550)
Disagree	12.6 (669)
Strongly disagree	2.2 (117)
**On-going follow-up with your physician helps early detection or prevention of cancer**	
Strongly agree	42.0 (2231)
Agree	49.9 (2651)
Disagree	7.3 (388)
Strongly disagree	0.8 (43)

**Table 3 healthcare-11-02475-t003:** Primary factors for perceived risk for colorectal cancer (CRC).

Factors for High Self-Perceived Risk for CRC		Factors for Low Self-Perceived Risk for CRC	
Item	Total (%)	Item	Total (%)
Family history of CRC	33.5	Regular medical check-ups	38.7
In the age group associated with CRC diagnosis	31.8	A healthy, balanced diet	34.1
Increasing incidence of CRC	29.3	No family history of cancer	32.0
General susceptibility to the disease	21.7	Asymptomatic	25.5
Sedentary lifestyle, no regular exercise	16.8	Active lifestyle, regular exercise	18.3
Overweight or obese	15.1	Non-smoker	17.7
Smoker	9.3	Not worried about CRC	17.4
Baseline: self-perception of high risk	1618	Baseline: self-perception of low risk	3695

**Table 4 healthcare-11-02475-t004:** Primary reasons for FOBT uptake or non-uptake.

Reasons for FOBT Uptake		Reasons for FOBT Non-Uptake	
Item	Total (%)	Item	Total (%)
Ruling out the possibility of having the disease	68.2	Healthy and asymptomatic	40.0
Early detection leads to effective treatment in most cases	63.7	Did not receive an invitation letter	39.7
Age recommended for CRC screening	47.1	Neglect or forgetfulness	28.5
Easy to deliver and receive results	33.8	COVID restrictions prevented visits to primary care center	24.8
Simple and easy to use	33.8	Public healthcare is saturated	22.0
Baseline: FOBT uptake	3319	Baseline: Prior knowledge of FOBT and non-uptake	1561

**Table 5 healthcare-11-02475-t005:** Bivariate analysis of variables and FOBT uptake.

		FOBT Uptake (%)		% (*n*)	χ² (V)
		Yes	No		
Receipt of invitation letter to carry out FOBT	Yes	77.1	22.9	100 (3619)	1284.007 * (0.492 *)
	No	23.9	76.1	100 (1694)	
Prior knowledge of FOBT	Yes	69.9	30.1	100 (4525)	832.126 * (0.396 *)
	No	13.3	86.7	100 (788)	
Frequency of primary care visits	Once to several times a month	65.0	35.0	100 (303)	91.974 * (0.132 *)
	Every two or three months	65.7	34.3	100 (1165)	
	Every six months	65.4	34.6	100 (1667)	
	Once a year	62.8	37.2	100 (1552)	
	Every few years	50.3	49.7	100 (511)	
	Never	31.3	68.7	100 (115)	
Perception of routine medical Check-ups	It is not necessary to perform these tests if a person is healthy	44.8	55.2	100 (4859)	65.952 * (0.111 *)
	Even if a person is healthy, performing these tests is always advised	64.1	35.9	100 (453)	
Perception of the effectiveness of cancer screening tests	Strongly agree	66.9	33.1	100 (3078)	83.125 * (0.125 *)
	Agree	57.8	42.2	100 (2069)	
	Disagree	41.7	58.3	100 (144)	
	Strongly disagree	27.3	72.7	100 (22)	
Age	50–54	49.8	50.2	100 (1558)	163.438 * (0.175 *)
	55–59	64.2	35.8	100 (1441)	
	60–64	69.1	30.9	100 (1268)	
	65–69	70.9	29.1	100 (1046)	
Total		62.5	37.5	100 (5313)	

* *p* value < 0.01.

**Table 6 healthcare-11-02475-t006:** Variables included in the logistic regression model.

	OR	95% CI	*p*-Value
Receipt of invitation letter (Yes)	7.346	(6.218–8.68)	<0.001
Prior knowledge of FOBT (Yes)	6.322	(4.901–8.154)	<0.001
Frequency of primary care visits (Every 6 months or less)	1.714	(1.392–2.111)	<0.001
Perception of routine check-ups (Positive perception)	1.688	(1.322–2.155)	<0.001
Age (62 to 69)	1.524	(1.296–1.792)	<0.001
Age (57 to 61)	1.492	(1.263–1.764)	<0.001
Age (50 to 56)	1		
Self-perceived risk for CRC (High/Very high)	1.406	(1.114–1.774)	0.004
Perception of test effectiveness (Strongly agree)	1.356	(1.181–1.557)	<0.001
Lead a healthy life (Slightly healthy/Not healthy at all)	1.252	(1.04–1.507)	0.018
Region (Andalucía)	1		
Region (Aragon)	1.694	(1.102–2.605)	0.016
Region (Asturias)	1.218	(0.788–1.884)	0.375
Region (Baleares)	1.469	(0.905–2.384)	0.119
Region (Canarias)	3.344	(2.362–4.734)	<0.001
Region (Cantabria)	2.033	(1.105–3.74)	0.022
Region (C. La Mancha)	1.608	(1.123–2.303)	0.010
Region (C. Leon)	1.997	(1.44–2.769)	<0.001
Region (Cataluña)	1.882	(1.496–2.366)	<0.001
Region (Valencia)	1.85	(1.433–2.389)	<0.001
Region (Extremadura)	1.553	(0.978–2.466)	0.062
Region (Galicia)	2.643	(1.892–3.691)	<0.001
Region (Madrid)	1.526	(1.21–1.924)	<0.001
Region (Murcia)	1.635	(1.064–2.513)	0.025
Region (Navarra)	5.061	(2.379–10.766)	<0.001
Region (Basque Country)	4.548	(3.038–6.807)	<0.001
Region (Rioja)	1.67	(0.738–3.78)	0.218
Region (Ceuta)	0.526	(0.08–3.467)	0.505
Region (Melilla)	1.011	(0.188–5.441)	0.990
Constant	0.298		<0.001

## Data Availability

The data underlying this article are available in Zenodo at https://doi.org/10.5281/zenodo.7970372 and will be shared on reasonable request to the corresponding author.

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
