# Peer review of "Knowledge, Compliance, and Inequities in Colon Cancer Screening in Spain: An Exploratory Study"

_healthcare, 2023, doi:10.3390/healthcare11182475_

Round 1

Reviewer 1 Report

Overall summary

The authors conducted a study to evaluate the awareness, attitudes, and perceptions of Spanish men and women aged 50 to 69 toward test-based CRC screening programs.

General comments

The article, although it is not new, is interesting and has great clinical relevance. However, there are deficiencies which should be addressed before publication could be considered.

INTRODUCTION

o   Lines 62-65. Reference 21 is missing.

o   Lines 73-76. Authors must include some references which support these statements.

MATERIAL AND METHODS

o   Subsection “participants and sample”. This subsection is incomplete. For example, it is not possible to know how the sample was collected. Furthermore, it is necessary to include the bounce rate for participating.

o   Table 1. This information must be included in the section “results”, instead of in the subsection of “participants and sample”.

o   Subsection “measures”. i) The questionnaire should be included as supplementary material; ii) the reference corresponding to the qualitative study is missing (lines 100-101); iii) The content of the questionnaire was reviewed by a group of experts, but were psychometric properties analysed?

o   In the subsection “data analysis” the authors should include the significance level used.

RESULTS

This section needs deep changes. In general, information in the main text and tables is overlapping. Furthermore, sometimes the information in the text is redundant, which makes this section a bit boring. I would recommend revise and rewrite it. Following, I have included several examples:

o   Subsection 3.1 and table 2. Information in these 2 paragraphs and table 2 overlaps. Information in the text is redundant (eg. A total of 68.3% of the sample rated their health status as good or very good, and 137 31.7% as fair, bad, or very bad).

o   Lines 157-162 and Table 3. Information in this paragraph and Table 3 overlaps.

o   Lines 164-168. Again, the information is redundant. For example: if you say that “83.1% of the sample perceived CRC as having a high or very high incidence rate in the general population as compared to other types of cancer”, it is not necessary to say that the rest of the sample perceived it as low or very low.

o   Lines 174-179 and Table 4. Information in this paragraph and Table 4 overlaps.

On the other hand, from my point of view, it is better to include statistical information in the table 5 than in the text (lines 198-211). Furthermore, why the bivariate analysis regarding socio-demographic variables was not included in the table 5?

DISCUSSION

The authors should improve this section. In this section, the authors must not only describe and compare their results with that of other studies, but also try to interpretate their results and to explain to what may be due the differences found between this study and others. Furthermore, authors must say what is new, what this research contributes that was not already known.

On the other hand, limitations of the study are missing.

REFERENCES

The references list must be reviewed (see https://www.mdpi.com/journal/healthcare/instructions#references). For example, the volume of the reference #6 is 95 (not 138), the authors must separate the name of the journal and the year…

On the other hand, try to update the references. Some of them are very old (eg. #16-20).

Other comments

o  Please, pay attention to the decimal separator:

·       Line 162. Change “17,7” by “17.7”

·       Table 3: change “1.618” by “1,618” and “3.695” by “3,695”

·       Table 4. Change “3.319” by “3,319” and “1.561” by “1,561”

o   Lines 172. Change “an” by “a”

Minor editing of English language required

Reviewer 2 Report

Dear authors, I appreciated your submission on the interesting research topic about the multifactorial nature of the propensity to the Colon Cancer Screening.

In general, your work is well structured and easy to read. Introduction and discussion are well written.

I’ve some suggestions to improve it:

·       Lines 164-166 : I suggest to rephrase in a clearer way

·       Line 182: Among the respondents that underwent the FOBT, 99.6%...... I suggest to rephrase in a better way

·       Lines 186-187. ….of those who did not take the FOBT, 67.1% agreed or strongly agreed that this test does not ensure the possibility of CRC incidence in the future. “Ensure” ?

·       Bivariate analysis:

You have used the Cramer V statistics to estimate the association between each variable and the likelihood of performing the FOBT test. It is methodologically correct but leads to a partial information, only.

I think that screening propensity is a multifactorial aspect that cannot be understood by breaking it down into single factors. I suggest adding a logistic regression model where the FOBT compliance is the dependent variable and all the other variables act as covariates. So, you can estimate the association between the FOBT compliance and each factor also considering the effect of all the other factors at the same time. 5,313 individuals are a large sample allowing to include many covariates.

I hope my suggestions would be helpful and I sincerely believe the audiences will benefit greatly from this work.

Dear authors,

I found the introduction and Discussion sections well written but I suggest restructuring the English Language in the Results section to improve the quality of your messages.

Round 2

Reviewer 1 Report

Dear Authors,

thank you very much for improving the manuscript according to my comments. 

The spelling and grammar  have been improved.

Reviewer 2 Report

Dear authors, I find that the quality of your work has improved and your message is clearer.